

# First steps towards mitochondrial pan-genomics: detailed analysis of *Fusarium graminearum* mitogenomes

Balázs Brankovics[1,2,3], Tomasz Kulik[4], Jakub Sawicki[4], Katarzyna Bilska[4], Hao Zhang[5], G Sybren de Hoog[2,3], Theo AJ van der Lee[1], Cees Waalwijk[1] and Anne D. van Diepeningen[1,2]

[1] Wageningen Plant Research, Wageningen University & Research, Wageningen, Netherlands
[2] Westerdijk Fungal Biodiversity Institute, Utrecht, Netherlands
[3] Institute for Biodiversity and Ecosystem Dynamics, University of Amsterdam, Amsterdam, Netherlands
[4] Department of Botany and Nature Protection, University of Warmia and Mazury, Olsztyn, Poland
[5] State Key Laboratory for Biology of Plant Diseases and Insect Pests, Institute of Plant Protection, Chinese Academy of Agriculture Sciences, Beijing, P.R. China

Corresponding author
Balázs Brankovics,
balazs.brankovics@wur.nl

## ABSTRACT

There is a gradual shift from representing a species' genome by a single reference genome sequence to a pan-genome representation. Pan-genomes are the abstract representations of the genomes of all the strains that are present in the population or species. In this study, we employed a pan-genomic approach to analyze the intraspecific mitochondrial genome diversity of *Fusarium graminearum*. We present an improved reference mitochondrial genome for *F. graminearum* with an intron-exon annotation that was verified using RNA-seq data. Each of the 24 studied isolates had a distinct mitochondrial sequence. Length variation in the *F. graminearum* mitogenome was found to be largely due to variation of intron regions (99.98%). The ''intronless'' mitogenome length was found to be quite stable and could be informative when comparing species. The coding regions showed high conservation, while the variability of intergenic regions was highest. However, the most important variable parts are the intron regions, because they contain approximately half of the variable sites, make up more than half of the mitogenome, and show presence/absence variation. Furthermore, our analyses show that the mitogenome of *F. graminearum* is recombining, as was previously shown in *F. oxysporum*, indicating that mitogenome recombination is a common phenomenon in *Fusarium*. The majority of mitochondrial introns in *F. graminearum* belongs to group I introns, which are associated with homing endonuclease genes (HEGs). Mitochondrial introns containing HE genes may spread within populations through homing, where the endonuclease recognizes and cleaves the recognition site in the target gene. After cleavage of the ''host'' gene, it is replaced by the gene copy containing the intron with HEG. We propose to use introns unique to a population for tracking the spread of the given population, because introns can spread through vertical inheritance, recombination as well as via horizontal transfer. We demonstrate how pooled sequencing of strains can be used for mining mitogenome data. The usage of pooled sequencing offers a scalable solution for population analysis and for species level comparisons studies. This study may serve as a basis for future mitochondrial genome variability studies and representations.

## INTRODUCTION

One of the most ideal markers for monitoring the distribution and spread of populations is the mitochondrial genome (*Harrison, 1989*; *Taylor, 1986*). Mitochondrial genomes are relatively small and, therefore, can be studied in their entirety. Due to its high copy number within individual cells, the mitochondrial genome is easy to access. Furthermore, it has a simple organization that makes homologous regions easy to identify. Finally, in many fungal groups mitogenomes are inherited uniparentally (*Basse, 2010*), which reduces the chance of heterologous recombination.

Mitochondrial sequences have been used for resolving phylogenetic and evolutionary relationships between fungi at all taxonomic levels (*Liu et al., 2009*; *Avila-Adame et al., 2006*; *Fourie et al., 2013*). In 2003, the DNA barcoding initiative started, aiming at using a single marker for taxon identification. The marker that was selected was a mitochondrial gene, cytochrome c oxidase I—COI or *cox1* (*Hebert et al., 2003*). In *Fusarium* however, the use of *cox1* was abandoned as a barcoding region, because the frequent presence of introns in the gene made this region impractical for PCR amplification (*Gilmore et al., 2009*). The difficulty in obtaining mitochondrial sequence, due to introns, lead to a general shift of interest from the mitochondrial to the nuclear genomes in fungi. Next generation sequencing (NGS) and new analysis methods have resolved this issue by dispensing with the need for PCR amplification for extracting mitochondiral sequences (*Brankovics et al., 2016*). In addition, *de novo* assembly of mitochondrial sequences from NGS data is not confounded by the presence of nuclear mitochondrial DNA segments (NUMTs), while NUMTs are known to cause problems in PCR-based barcoding (*Song et al., 2008*).

*Fusarium graminearum* is the major causative agent of Fusarium head blight (FHB), a devastating disease of small grain cereals. Besides reducing yield, the fungus contaminates crops with mycotoxins such as trichothecenes and zearalenone, which pose a serious threat to food and feed safety (*Desjardins, 2006*). Population studies of *F. graminearum* showed that the populations are highly dynamic and several displacements have been reported (*Gale et al., 2007*; *Ward et al., 2008*). Monitoring these population shifts is important, as they may differ in virulence, fungicide resistance and/or mycotoxin profile (*Gale et al., 2007*; *Zhang et al., 2012*).

The mitochondrial genome of *F. graminearum* encodes all genes typically associated with mtDNAs of filamentous fungi: two rRNA coding genes, 14 protein coding genes and a large set of tRNA coding genes (*Al-Reedy et al., 2012*). In addition, a large open reading frame with unknown function (LV-uORF) was found, flanked by tRNA genes. The first comparative studies of mitochondrial genomes of *Fusarium* spp. have revealed that *F. graminearum* has a significantly larger mitogenome than *Fusarium* spp. belonging to other species complexes analyzed so far (*Fourie et al., 2013*; *Al-Reedy et al., 2012*). Intron variation within the FGSC has not yet been analyzed, but the mitogenomes of different

species within the *F. fujikuroi* species complex showed diversity in intron content based on the sequences of *F. circinatum*, *F. fujikuroi* and *F. verticillioides* (*Fourie et al., 2013*).

Most mitochondrial introns found in *Fusarium* are group I introns. These introns are self-splicing ribozymes, which frequently contain homing endonuclease genes (HEGs) (*Haugen, Simon & Bhattacharya, 2005*). The combination of intron and HEG forms a mobile element that is able to invade intronless copies of the "host" gene (*Haugen, Simon & Bhattacharya, 2005*), thereby enabling horizontal spread of the mobile element through the population. This mechanism is called homing, since the homing endonuclease recognizes a target site of 15–45 bp, which makes the insertion highly sequence specific (*Haugen, Simon & Bhattacharya, 2005*). A functional homing endonuclease is needed for the homing of the intron, but the intron may be retained as long as the self-splicing function of the intron is intact. Since the mitochondrial genes are crucial for the proper functioning of the cell, if an intron loses its ability to self-splice, then the intron is lost through precise excision (*Goddard & Burt, 1999*). This mechanism allows an intron to spread in populations to strains that do not possess the given intron. This dispersion does not require further recombination. The mechanism does not allow one haplotype of an intron to replace another one, since the horizontal transfer is mediated only by the cleavage of an intronless copy. Hence, the replacement of one haplotype by another one can only be explained either by recombination or by loss of the original intron and insertion of the new haplotype.

Pan-genomes are the abstract representation of the genomes of all the strains that are present in the population. The idea of pan-genome or supra-genome comes from bacterial genomics, and originated from the distributed genome hypothesis (DGH) (*Ehrlich, 2001*; *Tettelin et al., 2005*). According to the DGH, each strain within a population/species contains a subset of contingency genes from within the supra-genome (pan-genome), i.e., the supra-genome is distributed among many individual strains (*Ehrlich, 2001*; *Ehrlich, Hu & Post, 2004*). Pan-genome based analysis can be used to identify conserved, variable and strain specific regions within a group of genomes. Pan-genomes can be also employed to contrast two populations or two species.

In order to create a pan-genome for the mitogenome of *F. graminearum*, we have to better understand the nature and dynamics of the diversity in the mitochondrial genome of this organism. To accomplish this, a reliable reference has to be established as a basis for all comparative analyses. To this end, we resequenced the reference strain of *F. graminearum*, PH-1, assembled its mitochondrial genome, improved its annotation and validated the annotation using RNA-seq. Subsequently, this reference was used to study the SNP frequencies, intron distribution and sequence variability of the different regions of the mitogenome within the species, by analyzing a total of 24 strains, which were individually sequenced, representing a wide range of hosts and geographic origins. Finally, we evaluated the efficacy of using pooled sequencing in assessing the mitogenome sequence diversity within a sample. Pooled sequencing offers the possibility of analyzing populations directly from field samples.

## MATERIALS & METHODS

### Strains

Thirteen *F. graminearum* strains were sequenced individually on the Illumina MiSeq platform (Table 1). In addition, *F. graminearum* strain PH-1 (CBS 123657, NRRL 31084) was sequenced on the Illumina HiSeq platform both as a single strain and as part of a pooled set of five *F. graminearum* strains (Table 1). Besides the newly sequenced strains, the whole genome sequencing reads of ten *F. graminearum* isolates were downloaded from the SRA database of NCBI that were produced by other research groups (*Laurent et al., 2017*; *Wang et al., 2017*). The outgroup, *F. gerlachii* strain was sequenced for an earlier publication (*Kulik et al., 2016*). A detailed description of the fungal strains is given in Table 1.

### Sequencing

#### Illumina MiSeq

Whole genome libraries were prepared using the Nextera XT kit (Illumina, San Diego, CA, USA) from gDNA extracted from mycelium. The constructed libraries were sequenced on the Illumina MiSeq platform with 250 bp paired-end read, version 2. The fungal genomes were sequenced in a multiplexed format (6–7 samples per run), where an oligonucleotide index barcode was embedded within adapter sequences that were ligated to DNA fragments (*Smith et al., 2010*). Next, the sequence reads were de-multiplexed and filtered for low quality base calls, trimming all bases from 5′ and 3′ read ends with Phred scores <Q30.

#### Illumina HiSeq

For *F. graminearum* strain PH-1 (CBS 123657, NRRL 31084) a random sheared shotgun library was prepared using the NEXTflex ChIP-seq Library prep kit with adaptations for low input gDNA according to the manufacturer's protocol (Bioscientific). The library was loaded as (part of) one lane of an Illumina paired-end flowcell for cluster generation using a cBot. Sequencing was done on an Illumina HiSeq2000 instrument using 101, 7, 101 flow cycles for forward, index and reverse reads respectively. De-multiplexing of resulting data was carried out using the Casava 1.8 software. Sequencing reads have been uploaded to the European Nucleotide Archive (ENA) with the accession number PRJEB18592.

The same method was applied for the pooled sequencing with the adjustment that random sheared shotgun library was prepared by using equal amounts of genomic DNA extract from all five strains (Table 1). Sequencing reads have been uploaded to the European Nucleotide Archive (ENA) with the accession number PRJEB18596.

#### Third party sequencing data

Besides the sequencing data that we have generated, we also made use of sequencing data produced by other research groups that had been submitted to SRA (Sequencing Read Archive) databases. This included a dataset of SRA data of six strains isolated from France (PRJNA295638; *Laurent et al., 2017*), three strains from China (PRJNA296400; *Wang et al., 2017*) and one strain from Australia (PRJNA235346; *Gardiner, Stiller & Kazan, 2014*). The mitochondrial genome sequences for the strains sequenced by third parties are available

**Table 1  List of *Fusarium* strains analysed in this study**

| Species | Strain | Origin | Host | Year of isolation | Sequenced individually or in a pool |
|---|---|---|---|---|---|
| *F. graminearum* | CBS123657 (PH-1) NRRL31084 | USA | maize | 1996 | both |
| *F. graminearum* | CBS119173 | USA | wheat head | 2005 | individually |
| *F. graminearum* | CBS139513 | Argentina | barley | 2011 | individually |
| *F. graminearum* | CBS139514 | Argentina | barley | 2010 | individually |
| *F. graminearum* | CBS119799 | South Africa | wheat kernel | 1987 | individually |
| *F. graminearum* | CBS119800 | South Africa | maize | 1990 | individually |
| *F. graminearum* | CBS110263 | Iran | maize | 1968 | individually |
| *F. graminearum* | CBS123688 | Sweden | oats | unknown | individually |
| *F. graminearum* | CBS128539 | Belgium | wheat kernel | 2007 | individually |
| *F. graminearum* | CBS138561 | Poland | wheat kernel | 2010 | individually |
| *F. graminearum* | CBS138562 | Poland | wheat kernel | 2010 | individually |
| *F. graminearum* | CBS138563 | Poland | wheat kernel | 2003 | individually |
| *F. graminearum* | CBS104.09 | unknown | unknown | 1909 | individually |
| *F. graminearum* | CBS185.32 | unknown | maize | 1932 | individually |
| *F. graminearum* | CS3005 | Australia | barley | 2001 | individually |
| *F. graminearum* | HN9-1 | China | wheat | 2002 | individually |
| *F. graminearum* | HN-Z6 | China | wheat | 2012 | individually |
| *F. graminearum* | INRA-156 | France | wheat | 2001 | individually |
| *F. graminearum* | INRA-159 | France | wheat | 2001 | individually |
| *F. graminearum* | INRA-164 | France | wheat | 2002 | individually |
| *F. graminearum* | INRA-171 | France | wheat | 2001 | individually |
| *F. graminearum* | INRA-181 | France | wheat | 2002 | individually |
| *F. graminearum* | INRA-195 | France | wheat | 2002 | individually |
| *F. graminearum* | YL-1 | China | wheat | 2012 | individually |
| *F. graminearum* | bfb0999_1 | China | barley | 2005 | pooled |
| *F. graminearum* | 68D2 | Netherlands | wheat | 2001 | pooled |
| *F. graminearum* | CHG013 | China | maize | 2005 | pooled |
| *F. graminearum* | CHG157 | China | barley | 2005 | pooled |
| *F. gerlachii* | CBS123666 | USA | wheat head | 2000 | individually |

in the Third Party Annotation Section of the DDBJ/ENA/GenBank databases under the accession numbers TPA: BK010538 –BK010547

## Assembly

GRAbB was used with SPAdes assembler to reconstruct the mitogenome of the strains. GRAbB (*Brankovics et al., 2016*) was chosen because it is a wrapper program for iterative *de novo* assembly based on a reference sequence. SPAdes 3.8.1 (*Bankevich et al., 2012*; *Nurk et al., 2013*) assembler was used, since it offers good insight for the user into the relationship between nodes in the assembly graph and the relationship between nodes, contigs and

scaffolds. The mitochondrial genomes were assembled from NGS reads using GRAbB by specifying the mitogenome sequence of PH-1 strain (HG970331) as query sequence.

For each individually sequenced strain it was possible to resolve the assembly to a single circular sequence. When the GRAbB run finished for the strains that were pooled for sequencing, the final assembly graph was visualized using Bandage (*Wick et al., 2015*) and the assembly was resolved to two circular sequence variants to capture all the variation within the dataset (Text S1). For the first circular sequence, referred to as "short", the shorter alternative contigs were included in the path at each position where continuity was ambiguous. For the other sequence, referred to as "long", the longer alternatives were included. In this way, all the different sequence regions were represented at least once in the two sequences.

### Sequence annotation

The initial mitogenome annotations were done using MFannot (http://megasun.bch.umontreal.ca/cgi-bin/mfannot/mfannotInterface.pl) and were manually improved: annotation of tRNA genes was improved using tRNAscan-SE (*Pavesi et al., 1994*), annotation of protein-coding genes and the *rnl* gene was corrected by aligning intronless homologs to the genome. Intron encoded proteins were identified using NCBI's ORF Finder (http://www.ncbi.nlm.nih.gov/gorf/gorf.html) and annotated using InterPro (*Mitchell et al., 2015*) and CD-Search (*Marchler-Bauer & Bryant, 2004*). The annotated mitochondrial genome sequences are available under the following GenBank accession numbers: BK010538–BK010547, KP966550–KP966561, KR011238 and MH412632.

### Read mapping and SNP discovery

The mitogenome of *F. graminearum* strain PH-1 and the two mitogenome sequences obtained from the assembly of the pooled dataset were used as reference sequences for the read mapping and SNP discovery. The read mapping was done using *aln* and *sampe* subcommands of the Burrows-Wheeler Alignment tool (BWA-0.7.12-r1034) (*Li & Durbin, 2009*). SNP calling was done using SAMtools mpileup (1.3.1) with *-g* and *-f* flag and BCFtools call (1.3.1) with *-mv* flag (*Li et al., 2009*).

### Coverage analysis

Coverage of different regions was estimated by, first, mapping reads of the pooled dataset to the reference sequence using the *sampe* subcommand of the Burrows-Wheeler Alignment tool (BWA-0.7.12-r1034) (*Li & Durbin, 2009*). Then, read coverage was calculated using the *genomecov* command of bedtools v2.26.0. The following single copy nuclear protein coding genes were used to represent single copy nuclear regions: $\gamma$-actin (*act*), $\beta$-tubulin II (*tub2*), calmodulin (*cal*), 60S ribosomal protein L10 (*rpl10a*), the second largest subunit of DNA-dependent RNA polymerase II (*rpb2*), translation elongation factor 1$\alpha$ (*tef1a*), translation elongation factor 3 (*tef3*) and topoisomerase I (*top1*). The reference sequences were extracted from the genome of PH-1 (four chromosomes: HG970332, HG970333, HG970334, and HG970335). The nuclear mitochondrial DNA segment (NUMT) used for coverage comparison was identified during the assembly of the pooled data (see Text S1).

### Intron validation

The RNA-seq data for *F. graminearum* PH-1 was downloaded from NCBI's SRA database, accession number PRJNA239711 (*Zhao et al., 2014*). Read mapping was done by HISAT2 aligner (*Kim, Langmead & Salzberg, 2015*) by specifying putative intron positions. The intron position were validated based on the splice site output file and by examining the mapping SAM file produced by the aligner.

### Linear model

R was used for linear model analysis to test whether the intron variation is the main reason of mitochondrial genome length variation within the species. The linear model was the following:

$$y = x + c$$

where $y$ was the total length of the mitochondrial genome, $x$ was the length of the intron sequences and $c$ was the $y$-intercept (average intronless length of the mitochondrial genomes). The $R^2$ value obtained from linear model analysis specifies what percentage of the variation of the dependent value (mitogenome length) is explained by the variation in the independent value (intron length).

$$R^2 = 1 - \frac{SS_{residual}}{SS_{total}}$$

Residual sums of squares ($SS_{residual}$) and total sums of squares ($SS_{total}$) were calculated using the *deviance* function of R.

### Comparative sequence analysis

The nucleotide sequences were aligned using MUSCLE (*Edgar, 2004a*; *Edgar, 2004b*). Sequence variability of given regions was calculated by aligning the sequences. Then the number of characters with multiple character states was calculated and divided by the total number of characters in the alignment. This step was done using fasta_variability from the fasta_tools package (https://github.com/b-brankovics/fasta_tools).

### Detecting the presence of recombination

The intergenic regions were analyzed using the $\Phi_w$-test implemented in SplitsTree (*Bruen, Philippe & Bryant, 2006*) to detect whether there is recombination in the mitochondrial genome.

## RESULTS

### Mitochondrial genome of *F. graminearum*

The mitochondrial genomes of all 24 strains sequenced individually were assembled into single circular contigs. The re-sequencing of the mitochondrial genome of *F. graminearum* strain PH-1 revealed two SNPs compared to the most recent published mitogenome assembly (HG970331.1) of the strain that was based on next generation sequencing reads (*King et al., 2015*). The correction of these SNPs was supported by the fact that all the other strains contained the same two SNPs obtained in the new assembly of PH-1. The newly

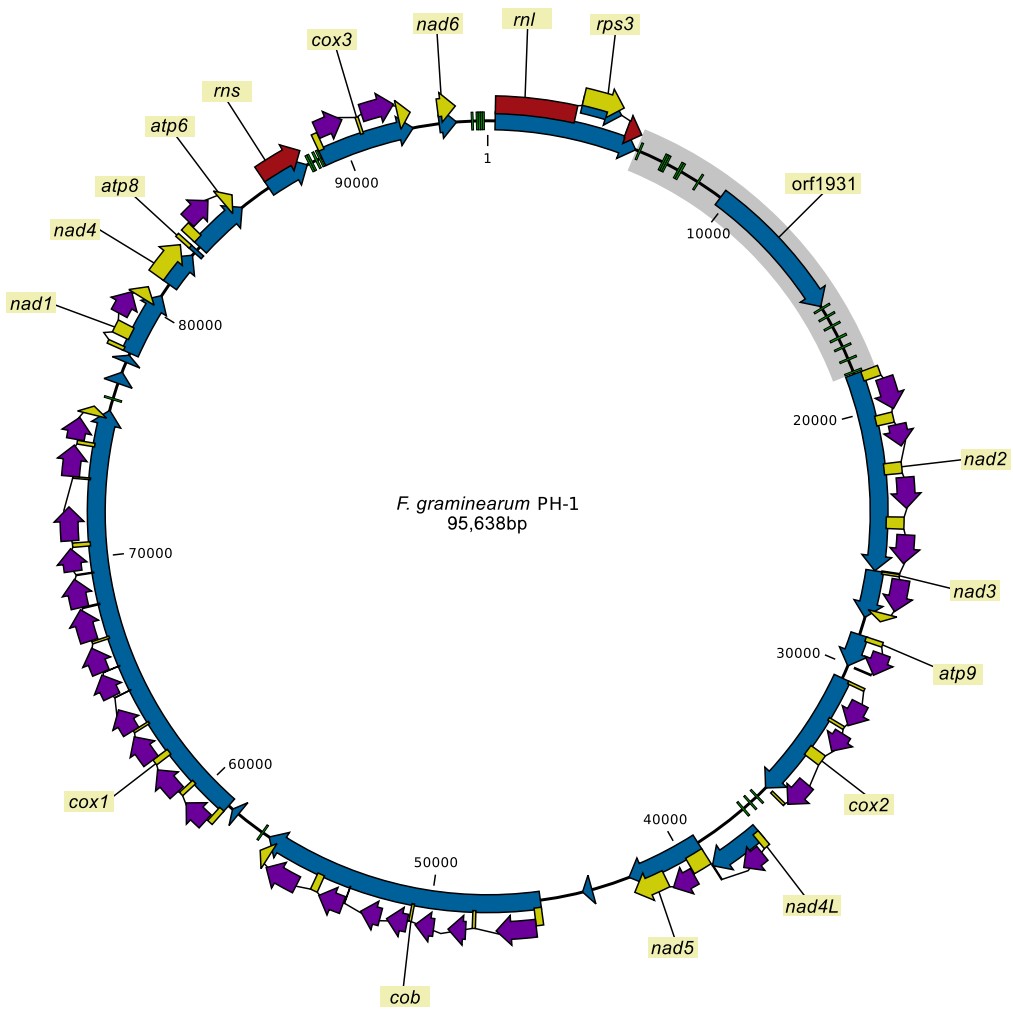

**Figure 1** **The mitogenome of *F. graminearum* strain PH-1.** Green blocks: tRNA coding genes, blue arrows: genes or ORFs (no labels added for short ORFs), yellow arrows: protein coding sequences, red arrows: rDNA coding sequence, purple arrows: intron encoded homing endonuclease genes, gray box: the large variable (LV) region with orf1931 (LV-uORF).

assembled mitochondrial genome of the PH-1 strain as well as the other mitochondrial genomes were annotated. The mitochondrial genomes of all strains contained the same set of genes in the same order and orientation (Fig. 1). To test whether the intron-exon models were predicted correctly, RNA-seq reads were mapped against the mitogenome of *F. graminearum* strain PH-1. The results of the read mapping supported all of the predicted intron-exon boundaries.

## Mitogenome variability in *F. graminearum*

The mitogenomes of *F. graminearum* strains analyzed showed variation in size, ranging from 93,560 bp to 101,424 bp (Table 2). To test whether intron variation is the main reason of mitochondrial genome length variation within the species, linear model analysis was

| Table 2 | Mitochondrial genome variation of the *Fusarium graminearum* strains. | | | | |
|---|---|---|---|---|---|
| **Strain ID** | **GenBank accession numbers** | **Size (bp)** | **Introns** | **Intronic (bp)** | **Core (bp)** |
| CBS123657 (PH-1) | MH412632 | 95,638 | 34 | 49,429 | 46,209 |
| CBS185.32 | KP966550 | 96,300 | 34 | 50,120 | 46,180 |
| CBS110263 | KP966551 | 97,364 | 35 | 51,165 | 46,199 |
| CBS119173 | KP966552 | 100,342 | 37 | 54,130 | 46,212 |
| CBS119799 | KP966553 | 96,005 | 35 | 49,919 | 46,086 |
| CBS119800 | KP966554 | 97,462 | 35 | 51,280 | 46,182 |
| CBS123688 | KP966555 | 95,035 | 34 | 48,837 | 46,198 |
| CBS128539 | KP966556 | 96,134 | 35 | 49,996 | 46,138 |
| CBS138561 | KP966557 | 95,034 | 34 | 48,837 | 46,197 |
| CBS138562 | KP966558 | 99,062 | 36 | 52,865 | 46,197 |
| CBS138563 | KP966559 | 99,068 | 36 | 52,865 | 46,203 |
| CBS139514 | KP966560 | 96,167 | 35 | 49,980 | 46,187 |
| CBS139513 | KP966561 | 95,041 | 34 | 48,837 | 46,204 |
| CBS104.09 | KR011238 | 97,460 | 35 | 51,280 | 46,180 |
| CS3005 | BK010538 | 93,560 | 33 | 47,381 | 46,179 |
| HN9-1 | BK010539 | 96,307 | 35 | 51,567 | 44740 |
| HN-Z6 | BK010540 | 97,767 | 34 | 50,120 | 47647 |
| INRA-156 | BK010541 | 101,424 | 37 | 55,243 | 46181 |
| INRA-159 | BK010542 | 96,199 | 35 | 49,996 | 46203 |
| INRA-164 | BK010543 | 99,678 | 37 | 53,476 | 46202 |
| INRA-171 | BK010544 | 96,199 | 35 | 49,996 | 46203 |
| INRA-181 | BK010545 | 96,187 | 35 | 49,996 | 46191 |
| INRA-195 | BK010546 | 97,358 | 35 | 51,165 | 46193 |
| YL-1 | BK010547 | 97,996 | 36 | 51,777 | 46219 |

Notes.
**Core** stands for the total mitogenome length minus the length of the intron regions.

used. The linear model that assumed that mitochondrial length variation is due only to variation of the length of intron regions explained 99.98% of intraspecific length variation observed in the data, showing that intron variation is the main reason behind intraspecific mitochondrial genome length variation. The standard deviation of the mitogenome length was 1,818 bp, which is 1.87% of the average mitochondrial genome length.

The coding regions (tRNA, rRNA and conserved protein coding genes) showed low levels of variation both within *F. graminearum* (0.02%) and when compared to *F. gerlachii* (0.02%). In addition, none of the SNPs found in protein coding regions caused amino acid substitution.

The large ORF with unknown function (LV-uORF) located in the large variable region of the mitogenome contained five SNPs within *F. graminearum* and the sequence in the *F. gerlachii* strain was identical to the most frequent haplotype within *F. graminearum*. All five SNPs resulted in amino acid substitution in the putative peptide sequences. The variability of the conserved protein coding regions was 0.02%, while the variability of the
**Table 3** Distribution of variation in the intron and intergenic regions within and between species.

| | Intraspecies | | | Interspecies | | |
|---|---|---|---|---|---|---|
| | Length (bp) | Variable positions | Variation frequency | Length (bp) | Variable positions | Variation frequency |
| Coding | 21,572 | 4 | 0.02% | 21,572 | 5 | 0.02% |
| Intron | 59,091 | 399 | 0.68% | 59,091 | 419 | 0.71% |
| Intergenic | 18,982 | 310 | 1.63% | 189,82 | 436 | 2.30% |

LV-uORF region was 0.09% within *F. graminearum*. The difference in variability was even more striking on the protein sequence level, where the conserved protein genes showed no variation, while the LV-uORF showed 0.26% variability.

The variability of the intergenic regions was 1.63% and 2.30% for intraspecies and interspecies, respectively. The overall sequence variability of intron sequences was 0.68% and 0.71% for intraspecies and interspecies, respectively. Although the variability of intron regions was significantly less than that of intergenic regions, both regions contained approximately equal numbers of variable sites (Table 3) due to the large length difference between the two regions. The intron regions were the most variable part of the mitochondrial genomes, because approximately half of the variable sites were located in introns, and introns were the only regions showing presence/absence variation within *F. graminearum*.

Interestingly, strains CBS 128539 and CBS 138561 had identical intergenic sequences, while strains CBS 104.09 and CBS 119800 (isolated 81 years apart) had identical intron sequences. However, each *F. graminearum* strain analyzed had a unique mitochondrial genome sequences.

## Intron patterns and recombination

A total of 39 intron sites were found in the individually sequenced dataset (Table S1). Out of the 39 introns, 32 were present in all strains and 21 of these showed no variation at the intraspecies level and 14 at the interspecies level. The introns that showed presence/absence variation within the dataset were cob-i159, cob-i201, cox1-i1287, cox2-i228, cox2-i318, cox2-i552 and nad2-i1632 (Fig. 2 and Table S1). The intron names contain the gene name where they are located and the coding nucleotide position of the host gene after which they were inserted.

It was not possible to group the strains based on their intron patterns (presence/absence for each intron) without allowing for multiple gain or loss of introns (Table S1). This could be the result of recombination of parental mitochondrial genomes or the horizontal transfer of introns. Recombination would affect all regions equally, while the horizontal transfer of introns by homing would affect mostly the intron sequences and their immediate vicinity. Recombination of the intergenic regions was well supported ($p = 2.26 * 10^{-6}$) by the $\Phi_w$-test.

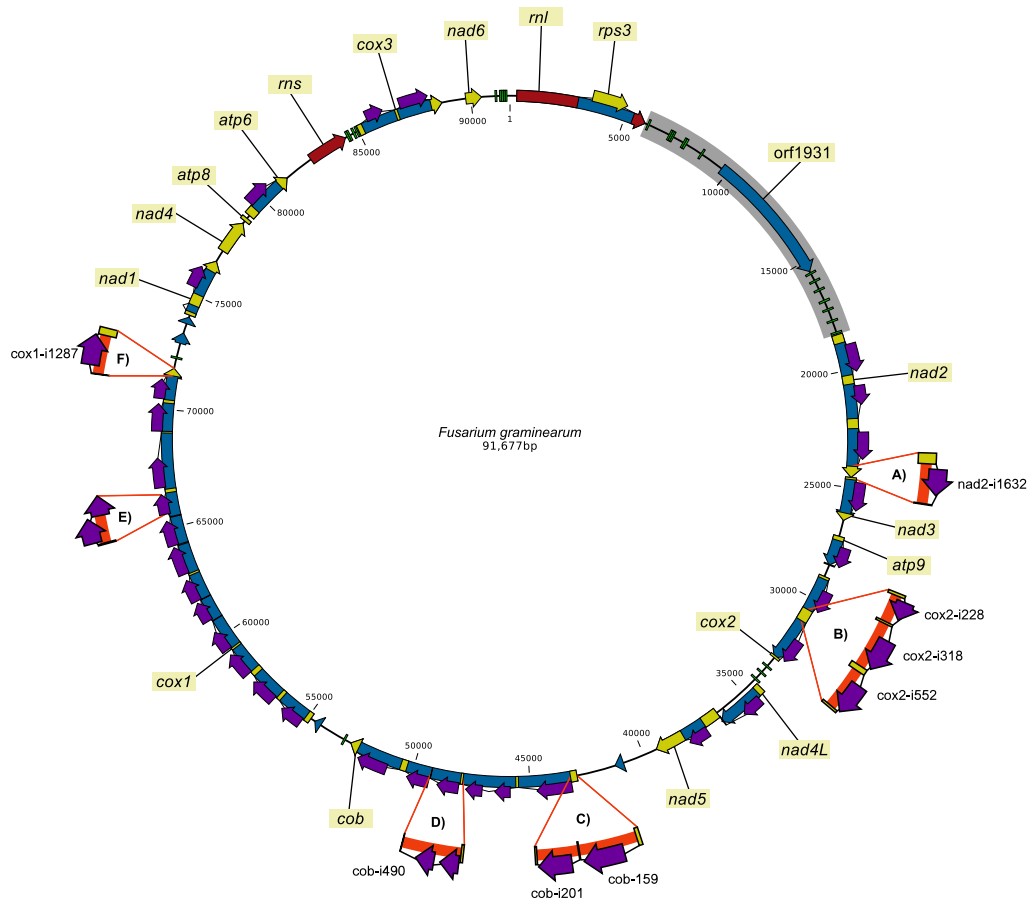

**Figure 2** **Pan-genomic representation of the presence/absence variation of introns in the mitochondrial genomes of the 24 *F. graminearum* strains.** In the figure, the thick orange lines highlight intron sequences in the alternative sequences. (SNPs and short indels are not indicated.) (A) The insertion of nad2-i1632; (B) the insertion of cox2-i228, cox2-i318 and cox2-i552; (C) the insertion of cob-i159 and cob-i201; (D) longer variant of cob-i490; (E) intron insertion in the HEG located in cox1-i906; and (F) the insertion of cox1-1287.

## Strategies to analyze pooled mitochondrial NGS data

Two approaches were used to explore the mitogenome variability in the pooled dataset: (i) assembling the reads *de novo* and (ii) mapping the reads to a reference sequence.

### De novo assembly approach

The assembly resulted in a graph that contained five ambiguous sites that represented four insertion/deletion variations (three intron presence/absence variations for cob-i201, cox1-i1287, cox2-i318, and a large insertion inside the cob-i490 intron) in the dataset, and one site (located in nad4L-i239) where two different alleles were found in the strain set (Text S1). These polymorphic sites were too far apart to establish linkage between them, so two alternative assemblies were extracted from the assembly graph: one with the shorter allele at all of the positions and one with the longer allele at all of the positions (Text S1). The

assembly method did not reveal SNP variations, only intron presence/absence variations and one replacement variation.

### Mapping approach

To assess the influence of the reference sequence on the mapping and SNP calling results, three sequences were used as reference in independent mapping runs: both of the sequences obtained from the assembly approach of the pooled dataset were used as references, beside the curated mitogenome of the PH-1 strain. Besides providing insight into the influence of the reference sequence to the downstream analysis, this also makes it possible to detect variation within intron sequences that may be absent in some of the reference sequences.

The lowest coverage detected for a single nucleotide allele was 21% of the reads that mapped to a given position. This is close to the expected value (20%) for an allele present in a single strain in a pool of five strains. This result showed that the method was sensitive enough to detect a SNP present in a single strain. Furthermore, the results of all three analyses identified the same polymorphic sites. This means that the choice of reference sequence did not influence the SNP detection results.

The three runs of read mapping and SNP calling revealed a total of 15 SNPs (Table 4). The allele ratios were identical even when the reference sequence used for the mapping was different, with one exception: position 90,636. At this position both PH-1 and the pooled assembly analysis showed 70% for the nucleotide present in the given reference and 30% for the alternative, despite the fact that the two references had different nucleotides at the given location (Table 4). Examination of the alignment of the reference sequences revealed that the sequence difference was not only a single nucleotide polymorphism at position 90,636, but there was a 8 bp long indel at position 90,627–90,634. This nearby indel influenced the mapping of reads containing the allele differing from the reference sequence. This was the reason why the SNP calling skewed in favor of the reference allele in both mappings.

### Coverage analysis

Coverage values were calculated for different genomic regions in order to determine whether coverage cutoffs could be used to differentiate between mitochondrial sequences and nuclear mitochondrial DNA segment (NUMT) sequences. The coverage of single copy nuclear regions that were present in all of the pooled strains was 290×. The coverage of the NUMT sequence was 230×, which suggests that it was present in four of the five pooled strains. The coverage of mitogenome regions that were present in all strains was 4,000×. While, the coverage of singleton mitochondrial regions, present only in a single strain, was 475×. The coverage gap was sufficiently large between shared single copy nuclear regions (290×) and singleton mitochondrial sequence (475×) to allow clear differentiation between them.

## DISCUSSION

Comparative genomics analyses are traditionally reference (*Laurent et al., 2017*) or pairwise based (*Fourie et al., 2013*). Reference based methods are efficient at identifying regions that

**Table 4** **List of single nucleotide polymorphisms identified in the pooled dataset of** *Fusarium graminearum* **strains.** Positions are aligned positions between the PH-1 reference sequence and the pooled sequences ("short" and "long"). "Reference" refers to the nucleotide found in the given reference sequence used for mapping, while "Alternative" refers to the nucleotide suggested by the mapped reads. Position 90,636 shows unusual ratios: in both mappings the reference nucleotide (C or A) has a frequency of 70% and the alternative nucleotide has 30%. This is due to an adjacent indel that affects the mapping results.

| Position | PH-1 | | Pooled | |
| --- | --- | --- | --- | --- |
| | Reference | Alternative | Reference | Alternative |
| 2,337 | A (0.77) | G (0.23) | A (0.77) | G (0.23) |
| 6,288 | C (0.41) | A (0.59) | A (0.61) | C (0.39) |
| 6,355 | T (0.42) | C (0.58) | C (0.60) | T (0.40) |
| 13,540 | C (0.78) | A (0.22) | C (0.78) | A (0.22) |
| 37,126 | C (0.75) | T (0.25) | C (0.75) | T (0.25) |
| 37,773 | A (0.75) | G (0.25) | A (0.75) | G (0.25) |
| 44,773 | A (0.62) | G (0.38) | A (0.62) | G (0.38) |
| 64,776 | G (0.53) | A (0.47) | G (0.53) | A (0.47) |
| 70,827 | A (0.62) | G (0.38) | A (0.62) | G (0.38) |
| 89,194 | G (0.57) | A (0.43) | G (0.57) | A (0.43) |
| 90,636 | C (0.70) | A (0.30) | A (0.70) | C (0.30) |
| 95,918 | A (0.43) | C (0.57) | C (0.59) | A (0.41) |
| 99,784 | A (0.40) | G (0.60) | G (0.62) | A (0.38) |
| 100,362 | C (0.42) | A (0.58) | A (0.59) | C (0.41) |
| 100,538 | G (0.42) | A (0.58) | A (0.61) | G (0.39) |

are present in the reference, but absent in other individuals, or detecting smaller variations, like SNPs. This method does not identify regions that are absent from the single reference, while these regions might be valuable for clustering the non-reference individuals. Pairwise comparison is able to identify unique regions for both individuals; however, it is difficult to scale to a larger sample size, because every individual has to be compared to every other individual, then the results have to be brought to the same scale.

To take full advantage of next generation sequencing data, a paradigm shift is needed: from focusing on a single reference genome to using a pan-genome, that is, a representation of all genomic content in a certain population, species or phylogenetic clade (*Computational Pan-Genomics Consortium, 2018*). In this study, we used an *ad hoc* pan-genomic analysis of the mitochondrial genomes of *Fusarium graminearum*. The reason for using an *ad hoc* approach is that pan-genomics is still a young field of research, and as such, there are no clear standards developed yet for analysis, for files or for data sharing. The goal of the analysis was to understand the nature and the dynamics of mitogenome variability, then to identify the implications of these results for mitogenome based population studies or track & trace implementations. The results of this study can be utilized for the development of suitable data structures and file formats for capturing the variability of mitochondrial pan-genomes.

In this study, we improved the mitochondrial genome reference for *F. graminearum* strain PH-1, which is recognized as the reference strain of this species for genomic studies

(*Al-Reedy et al., 2012*; *King et al., 2015*; *Cuomo et al., 2007*). The first mitochondrial genome sequence was produced using Sanger sequencing and primer walking by *Al-Reedy et al. (2012)*. The assembly was improved by *King et al. (2015)* using NGS reads. This assembly corrected 15 SNPs and 30 indels in the sequence. Here, we present a new assembly, which corrected two more SNPs, complete with a detailed annotation. The introns that were predicted during the annotation process were all verified by RNA-seq data.

The mitochondrial genomes of *F. graminearum* and *F. gerlachii* contained the same genes and ORFs in the same orientation. The coding sequences showed high levels of conservations, and all SNPs found in protein coding genes were synonymous substitutions. The genetic variation in the mitochondrial genome could be classified into two groups: small sequence variations (SNPs and short indels) and intron gain and loss. Although, variations resulting from SNPs and short indels were twice as frequent in intergenic regions as in intron regions, about half of the variable sites were located in intron regions. The second type of variation, the presence/absence of introns, accounted for 99.98% of the length variation between the mitochondrial genomes. In conclusion, the majority of the sequence variation within the species was related to intron regions: either SNPs and short indels or the presence/absence of complete introns. Thus, in mitogenome comparative analysis or pan-genomic studies, special attention should be given to accurately capturing the intron variation, since it is the most informative fraction of the mitogenome.

The annotation of strain CBS 119173 revealed a putative nested intron in cox1-i906. All other strains contain a haplotype that is 1,006 bp long, while this strain contains a haplotype that is 2,084 bp long. The sequence comparison indicates that the additional 1,078 bp region is an intron that was integrated inside the homing endonuclease of the acceptor intron. This putative intron contains an additional HEG, but the annotation pipeline did not identify the sequence as an intron. This type of construct is referred to as a twintron (*Copertino & Hallick, 1991*), and it shows that introns and intron encoded genes themselves are susceptible for intron invasions. The question is whether the invading intron has to retain its self-splicing function or the "host" (or acceptor) intron can splice the complete nested construct with its own self-splicing activity.

Most of the introns in *F. graminearum* are group I introns, and contain homing endonuclease genes (HEGs). Group I introns harboring a functional HEG can spread in a population through homing. Homing is facilitated by the homing endonuclease that cleaves the target gene at a 15–45 bp recognition site. The resulting double strand break stimulates homologous recombination based DNA repair. Since all copies of the mitochondrial genome that contain the recognition site are susceptible to the homing endonuclease, the only viable template for homologous repair is a genome that contains a copy of the intron. The insertion of the intron into the recognition site modifies the sequence, and it will no longer be recognized by the homing endonuclease.

The mitochondrial genome of *F. graminearum* shows evidence of heterologous recombination. We recently showed that mitochondrial recombination does also happen in the *F. oxysporum* species complex (*Brankovics et al., 2017*). Recombination of the mitochondrial genome in *Fusarium* appears to be a common phenomenon, since both *F. oxysporum* and *F. graminearum* show signs of mitochondrial recombination, despite

the fact that *F. oxysporum* is an asexual fungus with a putative parasexual cycle, while *F. graminearum* is a homothallic species that has an active sexual cycle (*Yun et al., 2000*). This finding does not prove that mitochondria are not uniparentally inherited in this fungus, but shows that heterologous recombination is widespread enough to be taken into consideration when using mitochondrial sequences for population level studies.

Based on the spreading mechanism of introns, introns could potentially be used for track and trace implementations, where introns could be viewed as "tags". The intron sequences spread through clonal & sexual reproduction, and through horizontal transfer. Due to the effect of the homing endonuclease, all offspring of a sexual cross would be "tagged" by all the introns that are specific to either parent. The appearance of a new intron in a population or the detection of an intron that is endemic to a region at a new geographic location could be signs of migration or gene flow.

An alternative way to sequencing strains individually is sequencing them in a pool. The pooled sequencing approach is more cost efficient than sequencing the strains separately. The data produced by pooled sequencing of strains from a given population could be viewed as the pan-genomic sequencing reads of that population. In this study, we have demonstrated how sequencing data from pools of strains can be mined for mitochondrial genome variation. Sequencing in pools has already been used to discover rare alleles of nuclear loci (*Raineri et al., 2012*). This method can be used for finding rare alleles, but it also allows a scalable solution for analyzing complete populations. So far, the application of pooled sequencing data has been used for SNP discovery in nuclear loci from samples (*Raineri et al., 2012*). However, analyzing mitochondrial genome data of fungi presents some additional challenges. We have demonstrated that besides SNPs, intron presence/absence variation is a major element of the mitogenome variation. To assess what kind of approach can detect intron presence/absence variation and SNP variation, we analyzed the data using a *de novo* assembly approach followed by a read mapping and SNP-calling approach. The results show that the assembly approach is able to identify sequence differences affecting sequence regions longer than individual sequencing reads, such as, insertions and deletions of intron sequence or long polymorphic sequences, while it is unable to identify SNPs or short indels. Read mapping and SNP calling analysis has to be performed to identify SNPs. This method in turn is unable to identify sequence differences affecting longer sequence regions. For optimal results, a sequential approach is needed for analyzing pooled samples: first, an assembly step to identify introns or larger indels absent from the reference genome, then using both the reference and the newly identified extra regions for read mapping and SNP-calling.

The disadvantages of pooled data are that short indel variation might be missed and linkage between markers is lost when using short read sequencing technologies, although linkage information is not crucial when comparing pan-genomes with each other. Furthermore, pooling large amount of strains could mean the loss of the coverage gap between mitochondrial copies and nuclear copies. This means that nuclear mitochondrial sequences (NUMTs) might affect the results by overestimating the amount of mitochondrial variation (*Song et al., 2008*). With sufficient caution the effects of NUMTs can be minimized, since they can be identified in the assembly step. In the assembly step, NUMTs

may appear as separate contigs, as in our example, or as new paths similar to introns with the significant difference that intron segments are joined to the rest of the mitochondrial assembly on both termini, while the flanking nuclear regions of NUMTs would only be joined on one of the termini of the segment. Despite these concerns, the benefit of pooled sequencing of large numbers of strains is that it offers a scalable solution for population or species level comparisons: after establishing a reference sequence, each population or species could then be represented by pan-genomes that are generated from pooled sequencing of multiple strains.

## CONCLUSIONS

We have improved the reference mitochondrial genome sequence for *F. graminearum*. Intraspecific mitochondrial genome length variations are mainly due to intron presence/absence variation, thus using ''intronless'' length—subtracting the length of the intron regions from the total mitogenome length—could be a valuable information when comparing species. Mitogenomes are also subject to recombination in both *F. graminearum* and in *F. oxysporum*, indicating that it is a common phenomenon in *Fusarium*. We proposed that introns unique to a single population could potentially be used to track the spread of the given population, because introns can spread through vertical inheritance, recombination and horizontal transfer. We also demonstrated how pooled sequencing of strains can be used for the mitogenome. The usage of pooled sequencing offers a scalable solution for population analysis and for species level comparisons. The results of this study represent an important step towards establishing pan-genomics for mitochondrial genomes.

### Funding

This work was supported by the Division for Earth and Life Sciences (ALW) with financial aid from the Netherlands Organization for Scientific Research (NWO, http://www.nwo.nl/) under grant number 833.13.006. The contributions of Cees Waalwijk were financially supported from the MycoKey project (Horizon2020, nr. 678781). The funders had no role in study design, data collection and analysis, decision to publish, or preparation of the manuscript.

### Grant Disclosures

The following grant information was disclosed by the authors:
Netherlands Organization for Scientific Research: 833.13.006.
MycoKey project Horizon2020: 678781.

### Competing Interests

The authors declare there are no competing interests.

### Author Contributions

- Balázs Brankovics conceived and designed the experiments, analyzed the data, prepared figures and/or tables, authored or reviewed drafts of the paper, approved the final draft.
- Tomasz Kulik conceived and designed the experiments, performed the experiments, analyzed the data, contributed reagents/materials/analysis tools, authored or reviewed drafts of the paper, approved the final draft.
- Jakub Sawicki and Katarzyna Bilska performed the experiments, authored or reviewed drafts of the paper, approved the final draft.
- Hao Zhang authored or reviewed drafts of the paper, approved the final draft.
- G Sybren de Hoog authored or reviewed drafts of the paper, approved the final draft.
- Theo A.J. van der Lee conceived and designed the experiments, performed the experiments, analyzed the data, contributed reagents/materials/analysis tools, authored or reviewed drafts of the paper, approved the final draft.
- Cees Waalwijk conceived and designed the experiments, performed the experiments, contributed reagents/materials/analysis tools, authored or reviewed drafts of the paper, approved the final draft.
- Anne D. van Diepeningen conceived and designed the experiments, performed the experiments, contributed reagents/materials/analysis tools, authored or reviewed drafts of the paper, approved the final draft.

### DNA Deposition

The following information was supplied regarding the deposition of DNA sequences:

The mitochondrial genome sequences described here are accessible via GenBank accession numbers KP966550–KP966561, KR011238, NC_025928, MH412632.

Mitochondrial sequence data for the strains sequenced by third party that were assembled and analyzed in this study are available in the Third Party Annotation Section of the DDBJ/ENA/GenBank databases under the accession numbers TPA: BK010538 –BK010547, and can also be found in the Supplemental Information.

### Data Availability

European Nucleotide Archive: PRJEB18592, PRJEB18596.

### Supplemental Information

Supplemental information for this article can be found online at http://dx.doi.org/10.7717/peerj.5963#supplemental-information.

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
