# Peer review of "First steps towards mitochondrial pan-genomics: detailed analysis of Fusarium graminearum mitogenomes"

_PeerJ, doi:10.7717/peerj.5963_

## Round 0.1 · original submission · Minor Revisions

The three reviewers see much merit in the manuscript and make a few suggestions. I agree with Reviewer 1 that some sections of the paper do not flow particularly well and would appreciate some effort to smooth out the text.

Reviewer 1 ·

Basic reporting

The approach and methods are well described and the work is thoroughly conducted. However, the narrative of the manuscript is often difficult to follow and disjointed in certain places.

Experimental design

The manuscript reports the use of a pan-genomic approach to analyze the mitochondrial genomes of 24 strains of Fusarium graminearum. The use of pan-genomes is novel approach for the analysis of mitochondrial genomes, and the manuscripts describes its effectiveness in assessing genome variability and the use of pooled sequencing.

Validity of the findings

The pan-genomic approach and use pools data sets has utility in the characterization of mitochondrial genomes, especially with the increase in NGS data. The sections of the paper that describe these approaches are well done, although some details of the mapping, coverage analysis, and coverage gaps are difficult to follow and would benefit by having additional descriptive statements. In contrast, the sections of the manuscript that focus on the use mitochondrial regions as barcodes for fungal populations fails to provide appropriate citations and lacks convincing evidence to support the claim that introns could be used to track the spread of a specific fungal population. The descriptions of group I intron mobility, in general, lacked important details and glossed over well-established concepts. The variation of group I intron content in fungal mitochondrial genomes is well documented, and there are numerous reports in the literature that high rates of transmission among fungal and plant mitogenomes. Thus the idea that there are ‘introns unique to a single population’ has not, to my knowledge, been substantiated. With the demonstrated ability of mobile group I introns to spread horizontally among strains of different vegetative compatibilities groups and even different genera (and on rare occasions, different kingdoms), it cannot be presumed that a natural population would have a specific set mitochondrial introns, let alone that an individual intron would be an accurate marker to track their spread. The manuscript would be greatly enhanced if it were primarily focused on the pros and cons of the pan-genome approach to analyze mitogenomes and potential use of pooled DNAs and remove or minimize the sections that discuss the possible use of introns as markers for a population.
Although the authors point out the limitations of using pooled data sets to identify SNPs, short indels, and linkage between markers, they underestimate the importance of retaining this information in understanding the origin and spread of specific mobile (and immobile) introns. The influence of mobile introns on genetic diversity of fungal mitogenomes goes well beyond their presence or absence. The co-conversion of SNPs flanking the insertion site associated with intron homing is a major contributor to the mtDNA diversity. The polymorphisms flanking intron insertion sites are generated via homologous recombination between different haplotypes or by error-prone repair mechanisms. The use of pooled data sets would not only have limited utility in characterizing these genetic differences, it would also potentially complicate the identification of the extant origin of a mobile intron.

Additional comments

Minor Comments
Abstract:
The term ‘recombination’ needs be defined. The authors seem to apply the term to describe recombination between mitochondrial genomes, and make no mention that homologous recombination is inherent to intron mobility. The term needs to specifically defined throughout.
The sentence stating that the ‘mitogenome of F. graminearum is recombining… indicating that the mitogenome recombination is a common phenomenon in Fusarium” is unnecessary. I cannot think of a fungal genus (nor any other eukaryote outside of metazoans) in which mitogenomes are not recombining, and this statement makes it seem as if this is a something in question, which it is not.
Line 46: Other citations to studies in which mitochondrial genomes are used to monitor species distribution are need. The Harrion, 1989 reference refers to animal genomes that have notoriously low levels of recombination. In addition, it would be worthwhile to point out the major challenges in applying one approach to all mitogonomes due the exceptional number of differences in size, diversity, and mechanisms of variation across kingdoms.
Line 50: The sentence is misleading; successful genotypes reflect the compatibility of nuclear and mitochondrial genomes, not just nuclear genotypes.
Line 58: The sentence is misleading; the use of mito genes as barcodes have mostly been abandoned in phylogenetic studies of fungal species.
Lines 194-196: Linear Model. It doesn’t seem necessary to include a model to quantify something when ‘percent intron’ would suffice (although, admittedly, this is first instance this reviewer has seen this metric applied).
Line 254: “However, all the strains had a unique mitochondrial genome sequence”. Vague statement. Is this referring to a single region?
Lines 256, 260 & 263: “Supplementary Table ” # missing in parentheses.
Lines 263-267: The explanation is too simplistic, and the statements should be removed or expanded to include a more complete description of the evolutionary events that could have produced the variability. Again, the meaning of the term ‘recombination’ is inadequately described. Intron homing could affect some intergenic regions, as the extent of flanking site co-conversion can exceed the length of exons.
Lines 367-377: I found this section to be difficult to follow. Why would pooling data reduce coverage of mito or nuclear copies? Why invoke PCR methods? How would NUMTs affect the results? The next sentence is very convoluted and difficult to understand. The final sentence of the paragraph also needs further explanation.
Lines 386- 393: See previous comment concerning the ubiquitous nature of recombination in fungal mitogenomes.
Lines 393-395: See previous comments and use of introns to track strains or populations
Lines 395-396: What do you mean by ‘tagged’?
Lines 396-397: This seems to imply that you would have to track a population over time to see the appearance of a new intron. Not sure how this would be accomplished.
Lines 400-405: What is being described is a ‘twintron’. These have been reported in numerous lineages in papers dating back to the early 1990s. Modify description and add citations.
Lines 406-407: This sentence should be removed, unless it can be further substantiated.
Lines 409-419: The Conclusion section should be re-written to focus on the novel approaches used to analyze the mitogenome of F. graminearum. The statements about the use of introns to track populations should be removed.

·

Basic reporting

The authors present a well written document in clear and easily understandable English. A sufficient amount of background information is provided in the Introduction, which also clarifies the context and scope of the study. Some small corrections are suggested in the attachment.
Some minor work is needed in the reference list. Some journals are abbreviated, others not, and some journals are both abbreviated and unabbreviated.
The document conforms to the traditional and professional structure of a scientific article, and the tables and figures are of publication quality. All raw data have been made available on public databases.

Experimental design

The sampling (i.e., isolates) was sufficient to support the assertions in the manuscript, and so were the in silico analyses.

Validity of the findings

The findings are valid for a selected assemblage of isolates from Fusarium graminearum. However, the results are exciting and should in be replicated for other Fusarium species. The potential use of mt-introns to track the movement of fungal pathogens around the world is a particularly exciting suggestion, and other groups should start investigating the effectivity and cost-effectiveness of such an approach.

Reviewer 3 ·

Basic reporting

no comment

Experimental design

no comment

Validity of the findings

no comment

Additional comments

To the editor of PeerJ
The manuscript “First steps towards mitochondrial pan-genomics: Detailed analysis of Fusarium graminearum mitogenomes” is a research paper that presents an improved version of the F. graminearum mitochondrial genomes and a paper that describes the use of pooled sequencing in assessing sequence diversity. The manuscript included sufficient background information and discussion however a minor revision is suggested before the manuscript can be published
L58: please clarify how NGS and new analysis methods have resolved the issue of cox1 and barcoding? Yes, NGS allow for more genomes to be sequenced and for populations to be analysed, however not sure if this will replace DNA barcoding?
L264-267: recombination should be a separate section and should be discussed in more detail.
Discussion general comments: In order to improve the overall flow of the discussion, the authors should follow the layout of the conclusion. Therefore, start with a discussion about the genome, the introns, recombination and all, and then have a section that describes the sequencing methods.

---

## Round 0.2 · accepted · Accept

Thank you for your attention to the reviewers' comments.

#